# Detailed Metabolic Characterization of Flowers and Hips of *Rosa gallica* L. Grown in Open Nature

**DOI:** 10.3390/plants12162979

**Published:** 2023-08-18

**Authors:** Nina Kunc, Metka Hudina, Maja Mikulič-Petkovšek, Jože Bavcon, Blanka Ravnjak, Gregor Osterc

**Affiliations:** 1Department of Agronomy, Biotechnical Faculty, University of Ljubljana, Jamnikarjeva 101, 1000 Ljubljana, Slovenia; 2University Botanic Garden, Biotechnical Faculty, University of Ljubljana, Ižanska cesta 15, 1000 Ljubljana, Slovenia

**Keywords:** *Rosa gallica*, *Rosa subcanina*, bioactive compounds, petals, rosehips, HPLC, secondary metabolites, primary metabolites

## Abstract

Our research aimed to investigate the primary and secondary metabolites of rosehips and petals of *R. gallica* in comparison with *R. subcanina*. *R. gallica* was chosen because it is still unexplored in terms of various bioactive substances and is strongly present in Slovenia. Given that roses are generally very variable and unstudied, our research will contribute to greater transparency and knowledge of the bioactive composition of rosehips and petals. We found a strong positive correlation between the total content of phenolics and ascorbic acid, between the total content of organic acids and the total content of carotenoids, and between the total content of sugars and the total content of organic acids. Hips of *R. gallica* contained higher amounts of sugars, ascorbic acid, and carotenoids than *R. subcanina*. Based on the composition of phenolic compounds in the petals, it is possible to distinguish between the two species. Among all the phenolic compounds in the petals, both genotypes are richest in gallotannins, followed by flavonols. Among anthocyanins, cyanidin-3-glucoside was determined, the content of which was also higher in *R. gallica*. It can be concluded that the studied hips had an extremely low sugar content and, consequently, an extremely high organic acid content. The content of carotenoids in hips was in the lower range of the average content compared to data from the literature. By optimizing the harvesting time, we could obtain a higher content of carotenoids, which could potentially be used for industrial purposes. However, we found that the analyzed petals were a rich source of phenolic compounds, which benefit the human body and could be potentially used in the food and cosmetic industries.

## 1. Introduction

Roses belong to the genus *Rosa*, where considerable diversity exists within and between species. They are characterized by their ability to germinate everywhere and are perfectly adapted to pioneer conditions, where there are almost always extreme growing conditions. They also grow in climatic vegetation communities, except they occupy different places and may not be as lush there. As an ornamental plant, roses have been common throughout the world since time immemorial. Rosehips, a rich source of bioactive compounds beneficial to humans, are becoming increasingly prominent [1,2].

Functional foods or dietary supplements that protect humans from oxidative stress and many diseases have recently become increasingly popular. Roses (rosehips and petals) are used for various purposes, including protecting health and treating influenza, infections, inflammatory diseases, and chronic pain. In addition, they have beneficial effects on skincare and healing ulcers. They are also used in foods and beverages, such as tea, jams, and jellies. They have recently been used as an ingredient in probiotic drinks, yoghurts and soups as dietary supplements. They are known to be antioxidant, anti-inflammatory and antibacterial, to improve the immune system, and assist against respiratory, gastric, and intestinal problems. In relation to these properties, several main species of roses are essential. Among them, *R. gallica* is very interesting since it is the most widespread European species and the parent species for several cultivars [3,4].

Song et al. [5] demonstrated that *R. gallica* petal extract promoted skin whitening and anti-wrinkle effects by regulating intracellular signaling, supporting its usefulness in cosmetic products for skin whitening. In addition, Jo et al. [3] concluded that *R. gallica* is promising for use as a functional ingredient in developing antiaging nutraceuticals. Ueno et al. [6] studied the neuropsychological effects of a water-soluble extract of *R. gallica* in male mice exposed to chronic stress and those under normal conditions. When the mice were exposed to stress, *R. gallica* had an anti-stress effect. They concluded that *R. gallica* has the potential as a medicinal plant that prevents stress. The medicinal effects of *R. subcanina* are also widely known and confirmed. Their effects are similar to those of *R. canina*, about which we found several studies and which is accepted as a control species in the *Rosa* genus. Tumbas Šaponjac et al. [7] confirmed that vitamin C and flavonoids are responsible for the antioxidant effect of rosehip tea, while only polyphenolics contribute to its antiproliferative effect. Daels-Rakotoarison et al. [8] reported that *R. canina* extract positively affects the respiratory activity of neutrophils. Ashtiyani et al. [9] concluded that *R. canina* fruit extract protects against renal dysfunction, oxidative stress and histological damage.

Uggla et al. [10] found that the two main sugars in *R. dumalis* and *R. rubiginosa* were glucose and fructose. Demir et al. [11] studied the organic acid and sugar compositions of *R. gallica*, *R. canina*, *R. dumalis*, *R. gallica*, *R. dumalis* subsp. *boissieri* and *R. hirtissima.* They also determined glucose and fructose as the main sugars. The highest content was determined in *R. canina* and *R. gallica*. They concluded that there were no great differences in the content of analyzed sugars. Rosus et al. [12] reported the total sugar content in various genotypes of roses, including *R. subcanina*. They found great qualitative and quantitative variability among the analyzed genotypes. Adamczak et al. [13] studied the bioactive composition of *R. canina*, *R. dumalis*, *R. glauca*, *R. inodora*, *R. jundzillii*, *R. rubiginosa*, *R. sherardii*, *R. tomentosa*, *R. villosa* and *R. zalana*. They found that the citric acid content was, on average higher than the ascorbic acid content. They also confirmed that there is great variability among genotypes. Javanmard et al. [14] reported the ascorbic acid content of five wild native *R. canina* species from Iran. Demir et al. [11] determined the contents of citric, malic, and ascorbic acid, in addition to the mentioned sugars. Ascorbic acid was present in the highest concentrations in the hips and malic acid in the lowest concentrations. Statistically significant differences were observed among the samples. High content of carotenoids was noted by Hornero-Mendez and Minquez-Mosquera [15], who listed β-carotene and lycopene as the main carotenoids. This is also consistent with other reports [16,17]. Olsson et al. [18] concluded that rosehips have a high content of total carotenoids compared to other berries and small fruits. This was found by researching ten fruit species whose carotenoid content differed almost 150 times.

Kunc et al. [19] studied the phenolic profile of *R. pendulina*, *R. spinosissima* and their hybrid *R. pendulina* × *spinosissima* (*R. reversa*). They found that out of 28 different phenolic compounds identified, quercetin-3-glucuronide was only present in the petals of the hybrid. The highest content of total phenolics was found in *R. spinosissima*. Cunja et al. [20] reported the phenolic profile of petals of *Rosa canina*, *Rosa glauca*, *Rosa rubuginosa* and *Rosa sempervirens*, as well as three modern cultivars ‘Rosarium Uetersen’, ‘Ulrich Brunner Fils’ and ‘Schwansse’. They found seven different anthocyanins and thirty-one flavonols, as well as 14 phenolic acids and their derivatives, 15 flavonols and 20 tannins. Cendrovski et al. [21] reported that *R. rugosa* petals are a rich source of phenolic compounds, which determine their antioxidant properties. The main polyphenolics were ellagitannins, accounting for 69 to 74% of all petal polyphenolics. Four other anthocyanins were identified: cyanidin 3,5-di-O-glucoside, peonidin 3-O-sophoroside, peonidin 3,5-di-O-glucoside and peonidin 3-O-glucoside, of which peonidin 3,5-di-O-glucoside accounted for approximately 85% of all anthocyanin compounds identified. Cunja et al. [20] determined thirty-one flavonols in rose petals; their content varied widely among the species and cultivars studied. Shameh et al. [22] reported eight different phenolic acids in the petals of six *Rosa* species. They listed gallic acid, caffeic acid, chlorogenic acid, *p*-coumaric acid, rutin, apigenin, cinnamic acid and quercetin.

Our research aimed to investigate the primary and secondary metabolites of rosehips and petals of *R. gallica* compared to *R. subcanina*, which, although not as common as some other wild roses, is also found in Slovenia. *R. gallica* was chosen because it is still unexplored in various bioactive substances and is strongly present in Slovenia. Our analyses are based on the research of Kunc et al. [1], who analyzed a broad spectrum of different bioactive compounds in roses. Considering that roses, in general, are highly variable and unexplored, our research will contribute to greater knowledge of the bioactive composition of rosehip fruits and flowers and encourage researchers to analyze the mentioned plants further. By comparing our results with those of other research, we can predict whether the petals and hips of *R. gallica*, compared to *R. subcanina*, are a rich source of bioactive compounds that could potentially be used in the food industry, alternative treatments, and cosmetic preparations.

## 2. Results

### 2.1. Primary and Secondary Metabolites in Rosehips

The total sugar content (sucrose, glucose, fructose and alcoholic sugars sorbitol and mannitol) is shown in Table 1. It can be seen that there is no statistically significant difference between the total sugar content of *R. gallica* and that of *R. subcanina*. The dominant sugar was fructose, followed by sucrose in *R. gallica* and glucose in *R. subcanina*. Sorbitol was present only in *R. gallica*. The differences between the individual sugars were also not significant.

The total organic acids content (Table 2) was significantly higher in the hips of *R. subcanina* than in the hips of *R. gallica*. The hips of both genotypes had the highest content of quinic acid, the hips of *R. gallica* 44.07 g/kg FW and the hips of *R. subcanina* 55.11 g/kg FW. The second highest content in the observed hips was citric acid. Shikimic acid and fumaric acid were present in lower amounts. The fumaric acid content was slightly higher in *R. subcanina* than in *R. gallica*.

The content of ascorbic acid (Figure 1) was significantly higher in the hips of *R. gallica* (5.39 g/kg FW) than in the hips of *R. subcanina* (1.17 g/kg FW).

There were no significant differences in the values of total carotenoid content in the hips of *R. gallica*, with 32.39 mg/100 g FW and in those of *R. subcanina*, with 28.27 mg/100 g FW (Table 3).

For all analyzed carotenoids (Table 3), there were no statistically significant differences between the genotypes studied. The predominant carotenoid was ẞ-carotene, with 27.56 mg/100 g FW in the hips of *R. gallica* compared to the hips of *R. subcanina*, with 23.37 mg/100 g FW, followed by lycopene. The content of α-carotene was higher in *R. gallica* samples. The contents of zeaxanthin and lutein were higher in *R. subcanina*.

The total content of phenolic compounds in the pulp with skin was significantly higher in the hips of *R. gallica* (15,767.21 mg/kg FW) than in the hips of *R. subcanina* (5305.45 mg/kg FW) (Table 4). However, there was also a statistically significant difference between the total content of phenolic compounds in the seeds of the studied genotypes. The seeds of *R. gallica* had a slightly higher content (1711.60 mg/kg FW) of phenolic compounds than those of *R. subcanina* (1263.08 mg/kg FW). The main phenolic groups determined in the pulp with skin and in the seeds of hips of the studied rose genotypes were hydroxybenzoic acid derivatives (HBA), hydroxycinnamic acid derivatives (HCA), gallotannins, ellagitannins, flavanols and flavonols. Flavones were present only in the pulp with skin. Dihydrochalcones were detected only in the seeds. In *R. subcanina*, flavanols predominated in the pulp with skin (3608.40 mg/kg FW) and the seeds (483.07 mg/kg FW). In *R. gallica*, flavanols were the dominant group in the pulp with skin, with 12,293.14 mg/kg FW, while dihydrochalcone was predominant in the seeds, with 1098.45 mg/kg FW. Cyanidin-3-glucoside was also identified in the pulp with skin of *R. subcanina* and in the seeds of *R. gallica*.

Based on the correlation test, a strong positive correlation was found between total phenolics content (TPC) and ascorbic acid (AC), between total organic acids content (TOC) and total carotenoids content (TCC) and between total sugar content (TSC) and TOC (Table 5 and Figure 2).

Principal component analysis (PCA) of all samples and metabolites was performed to provide a comprehensive picture of the analyses of rosehips (pulp with peel) in our study (Figure 3). PCA showed that two major components characterized TSC, TOC, AC, TCC, and TPC in rosehips (pulp with skin) of *R. gallica* and *R. subcanina*. The first and second components of the PCA model for the total data accounted for 81.7% (44.2% and 37.5%, respectively) of the total variance. It can be seen that the samples belonging to the same genotype are close to each other. This means the substances used to distinguish between the two species were appropriately used. The TPC and AC analyses significantly describe the samples of *R. gallica*, while the TCC and TOC analyses better describe the samples of *R. subcanina*. TSC analyses are not significant for either *R. gallica* or *R. subcanina* samples.

### 2.2. Bioactive Compounds in Rose Petals

The total content of phenolic compounds in rose petals (Figure 4) was higher in *R. gallica* (46,891.42 mg/kg FW) than in *R. subcanina* (35,514.02 mg/kg FW).

The content of hydroxycinnamic acid (HCA) derivatives (Table 6) was higher in *R. gallica* (73.87 mg/kg FW) than in *R. subcanina* (19.56 mg/kg FW). Caffeoylquinic acid was present only in the petals of *R. gallica*. There was a statistically significant difference between the samples studied. The content of gallotannins was higher in *R. subcanina*, 28,519.34 mg/kg FW. In *R. gallica*, the content of gallotannins reached 18,838.37 mg/kg FW. The predominant gallotanin was trigalloyl hexoside 1. There was a statistically significant difference between the total content of gallotonins. There is also a statistically significant difference between the content of ellagitannins in our samples. Their content in *R. gallica* was 884.29 mg/kg FW and in *R. subcanina* 165.23 mg/kg FW.

There was a statistically significant difference between the total content of flavanols and flavonols in the rose petals (Table 7). The content of flavanols was 3566.61 mg/kg FW in the petals of *R. gallica* and 1160.07 mg/kg FW in the petals of *R. subcanina*. The flavanol with the highest content in both studied roses was the dimer PA monogallate 1. The lowest value was found for epicatechin. The content of flavonols was significantly higher in the petals of *R. gallica*, 22,260.95 mg/kg FW, while in *R. subcanina,* it was only 5609.20 mg/kg FW. Petals of *R. gallica* had the highest content of quercetin dihexoside 2 (6624.44 mg/kg FW), which was not present in the petals of *R. subcanina*—in petals of *R. subcanina*, quercetin-3-arabinofuranoside dominated, with a content of 2541.73 mg/kg FW. Dihydrochalcone phloridzin was found only in the petals of *R. gallica*, in which its value reached 1084.44 mg/kg FW.

Total anthocyanin content was determined using cyanidin-3-glucoside as the chemical standard in the petals of *R. gallica* and *R. subcanina* (Table 8). A statistically significant difference existed between the average content of analyzed anthocyanin in the genotypes studied. *R. gallica* had a higher content than *R. subcanina*.

## 3. Discussion

We determined the content of primary and secondary metabolites in the hips and petals of the not yet well-studied rose genotype *R. gallica*, which grows naturally in the southwestern part of Slovenia, in Podgorje. We compared the data with the results measured in the control genotype *R. subcanina*, which grows in the same area, close to *R. gallica*. Five different types of sugars were determined in the hips: sucrose, glucose, fructose and the sugar alcohols sorbitol and mannitol. Sorbitol was present only in *R. gallica*. The total sugar content in the hips of *R. gallica* was 9.49 g/kg FW, whereas, in *R. subcanina,* it was 6.12 g/kg FW. Yoruk et al. [23] determined the sugar content of *R. iberica*, *R. canina*, *R. villosa*, *R. dumalis* and *R. pisiformis*. Fructose content ranged from 13.58 to 18.44 g/kg FW, glucose from 6.89 to 10.04 g/kg FW and sucrose from 0.57 to 5.61 g/kg FW. The content of all sugars in our experiment was much lower than in the experiment of Yoruk et al. [23]. The sucrose content of our samples was within the range reported by Yoruk et al. [23]. It was the sugar with the lowest content. As Yoruk et al. [23] noted, such differences in content are due to environmental conditions such as climate, soil structure and plant genetics. When plants are exposed to drought stress, solid compounds, especially sugars, are synthesised to regulate osmotic potential [24]. Similar results were also reported by Abaci et al. [25], who studied the sugar content of *Rosa iberica* Stev. The total content was 267.4 g/kg FW, with the major sugar being glucose, followed by fructose, sorbitol, and sucrose. Their results also showed extremely high values of total sugar content compared to ours. The rosehips investigated by Yoruk et al. [23] and Abaci et al. [25] were grown in Turkey, in the area of Lake Van [23] and are widespread in the north and east Anatolia [25], where there are completely different conditions for growth compared to the rosehips that we included in the research. We assume that rosehips in Turkey were extremely more exposed to stress conditions and that the lack of water in 2021 [1] in our area was so small compared to the Turkish stress conditions that it did not cause such an intense increase in sugars. Similarly, high values were also reported by Demir et al. [11] and Rosu et al. [12]. If the results are compared with those of the research conducted by Cunja et al. [26], who analyzed samples of *R. canina*, which also grew in Slovenia, it can be seen that the results are quite similar. For easier comparison, we converted our FW values to dry weight (DW), which is 37% FW in our case. The total sugar content was thus 2.26 and 3.5 g/kg DW. Cunja et al. [26] reported that the total sugar contents of *R. canina* were between 0.26 and 0.48 g/kg DW. Our samples thus had higher sugar contents. The increase in sugar content may be due to higher altitude and greater drought stress than the samples studied by Cunja et al. [26]. It should be noted that the mentioned research involved different genotypes, which additionally contributes to the different sugar contents.

Organic acids were citric, malic, quinic, shikimic and fumaric. It can be seen that the total acid content in our samples was higher than that in the literature. Cunja et al. [26] reported that citric acid was the most important organic acid detected in rosehips (up to 58% of the total organic acids). In our samples, we found a higher content of quinic acid and a lower one of citric acid. However, citric acid was immediately after quinic acid. Peña et al. [27] reported that the total organic acid content of *R. canina* and *R. rubiginosa* ranged from 46.2 g/kg to 73.2 g/kg FW. Okatan et al. [28] reported that the citric acid content in their samples of *R. canina* was between 15.9 and 22 g/kg FW. Quinic acid, the most available acid in our experiment (44.07 and 55.11 g/kg FW), ranged from 48 to 72 g/kg FW in their samples. Quinic acid was also the most available organic acid in the hybrid *R. pendulina * × *spinosissima* (48.8 g/kg FW), while the content of this acid in the original species *R. pendulina * was only 0.29 g/kg FW in the experiment mentioned by Kunc et al. [29]. As reported by Wang et al. [30], the main indicator of fruit quality is the content and ratio between soluble sugars and organic acids. Akagić et al. [31] reported that the ratio between sugars and acids strongly influences the fruit’s taste, shelf life and nutritional properties and are reliable indicators of consumer acceptability. In addition, changes in the content and composition of sugars and organic acids are also reflected in changes in the quality of fresh fruit and its products. Sugars also participate in the biosynthesis of polyphenolics, which means that the more sugar some fruit contains, the higher the polyphenol content is essential because of the nutrients that polyphenolics add to food. For an easier idea of how sweet or sour the fruits we analyzed were, we compared the results with those reported by Mahmood et al. [32]. Namely, they found that the total sugar content (sum of sucrose, glucose, and fructose) for ripe strawberries was from 48 to 54 g/kg FW, and for mulberries, the values ranged from 79.3 g/kg to 143.9 g/kg FW. The total content of organic acids in the studied strawberries varied between 13 and 16 g/kg. Mikulic-Petkovsek et al. [33] reported that the organic acid content of 20 grape varieties varied between 5.8 and 10.8 g/kg FW. The total sugar content of blueberries ranged from 49.99 to 111.67 g/kg FW, and the organic acid content ranged from 2.50 to 14.23 g/kg FW [34].

The content of ascorbic acid Is highly dependent on a variety of factors. The most Important are cultivar, stage of ripeness, altitude etc. The content of ascorbic acid in our samples was 5.39 g/kg FW for *R. gallica* and 1.17 g/kg FW for *R. subcanina*. Comparing our results with the results of an experiment conducted by Rosus et al. [12], we note that the mentioned authors determined higher contents of ascorbic acid, from 8.7 g/kg FW in *R. rubiginosa* to 6.2 g/kg FW in *R. caesia*. Alp et al. [35] determined the ascorbic acid content of rosehips from 10 different wild *R. dumalis* genotypes. The content ranged from 4.02 to 5.11 g/kg FW. In previous studies, the ascorbic acid content in rosehip fruits has been quite variable, with a range of 1.80 to 9.65 g/kg FW [36,37,38,39,40]. Krzaczek et al. [41] reported that the taxonomic assignment level below species plays a major role in ascorbic acid accumulation. Roman et al. [39] also studied ascorbic acid in samples of *R. canina* from Transylvania. They found that the ascorbic acid content of frozen samples grown at 1250 m above sea level was 3.6 g/kg, and of those grown at 440 m was 1.12 g/kg of frozen pulp. They found a good correlation between ascorbic acid content in rosehips and altitude. The site’s altitude in our experiment is about 500 m above sea level, and the ascorbic acid content is even lower than that found by Roman et al. [39]. Kunc et al. [29] reported that the content of ascorbic acid in the rosehips of *R. pendulina* × *spinosissima* was 10.45 g/kg FW, which is almost twice that in *R. penduline,* with 5.30 g/kg FW. Our experiment found that the ascorbic acid content of the two genotypes we studied was very similar to the content measured in *R. pendulina* in an experiment described by Kunc et al. [29]. Medveckiene et al. [42] studied five different rosehips, including *R. canina*, and found that the content of the above acid was 3.85 g/kg FW, the lowest of all the rosehips studied. The highest content (7.4 g/kg FW) was determined in *R. rugosa* ‘Rubra’.

Five carotenoids (lutein, zeaxanthin, lycopene, α-carotene, and ẞ-carotene) were determined in *R. gallica* and *R. subcanina samples*. Total carotenoid content was higher in *R. gallica* than in *R. subcanina*. Alp et al. [35] found that the total carotenoid content of 10 *R. dumalis* genotypes ranged from 47 to 85 mg/100 g FW. Rosus et al. [12] reported that the total carotenoid content of *R. subcanina* was 34.95 mg/100 g FW. In our samples, ẞ-carotene was present in the largest proportions (23.37 and 27.56 mg/100 g FW). Zhong et al. [43] studied the carotenoid content of *R. rubiginosa, R. multiflora, R. virginiana* and *R. rugosa*. They found a large difference in carotenoid composition among the roses. *R. virginiana* had the most diverse carotenoid composition and the highest carotenoid content. It was found that the group of lycopene together with ẞ-carotene accounted for 28 to 54% of the content of all carotenoids. Medveckiene et al. [42] reported that the content of ẞ-carotene ranged from 3.95 mg/100 g to 31.4 mg/100 g. In our samples, lycopene content was found to be 2.19 mg/100 g FW to 2.21 mg/100 g FW. The carotenoid with the lowest content in our samples was zeaxanthin, 0.29 to 0.33 mg/100 g FW. Similar contents were also reported by Medveckiene et al. [42], 0.23 to 0.32 mg/100 g. As can be seen from the above data, rosehips are a rich source of carotenoids, but there are differences in the content, which is the result of genetic variations, maturity level, agro-meteorological conditions, growing conditions, storage, and analysis method [44]. Because of all these factors, we assume that the content in our samples is also in the lower range of the average carotenoid content compared to data from the literature.

Total phenolics content in pulp with skin was extremely high in *R. gallica* (15.8 g/kg FW) compared to *R. subcanina* (5.31 g/kg FW). The total content of phenolic compounds in seeds was 1.27 g/kg FW in *R. subcanina* and 1.71 g/kg FW in *R. gallica*. Considering how low the sugar content determined in the samples was, the lower content of phenolic substances in the samples of *R. subcanina*, compared to *R. gallica*, is also logical. Flavones were determined only in the fruit pulp with skin and dihydrochalcones in the seeds. Najda and Buczkowska [45] investigated the phenolic content of 5 different rosehips. The phenolic content was between 1.1 g/kg FW and 2.2 g/kg FW. Pena et al. [27] reported the total phenolic content in *Rosa* spp. Varied between 0.3 g/kg and 14 g/kg. Unlike the aforementioned studies, much higher values are reported by Demir et al. [11], who found that the total content of phenolic compounds in *R. dumalis* subs. *Boissieri* was 52.94 g/kg and in *R. canina* 31.08 g/kg.

We determined 53 phenolic compounds in the petals of *R. gallica* compared to only 31 in *R. subcanina*. Cunja et al. [20], who studied *R. canina*, also determined 31 different phenolic compounds in this species, morphologically very similar to *R. subcanina*. Cendrowski et al. [21] determined 20 phenolic compounds in *R. rugosa* petals identified by UPLC-ESI-MS. Ellagitannins represented the major fraction, accounting for 69 to 74% of petal polyphenolics. In our samples, gallotannins dominated in *R. subcanina*, accounting for up to 81% of the phenolic compounds identified. In *R. gallica*, flavonols dominated, with a proportion of 47%, immediately followed by gallotannins with 40%. In contrast to Cendrowski et al. [21], Kumar et al. [46], Velioglu and Mazza [47] and Ochir et al. [48], we determined quercetin and kaempferol derivatives in our samples. Total phenolic compound content ranged from 22.08 g/kg FW (2012 season) to 25.90 g/kg FW (2013 season) in Cendrowski et al. [21]. In our studied samples, these values were 35.5 g/kg FW for *R. subcanina* and 46.9 g/kg FW for *R. gallica*. In comparison with the research results of Kunc et al. [1,19], who studied the phenolic composition of the flowers of *R. pendulina*, *R. spinosissima* and their hybrids, we found that the petals of *R. gallica* and *R. subcanina* had much lower values than those of the aforementioned roses. Of the flavonols, Cunja et al. [20] determined quercetin acetyl hexoside rhamnoside, kaempferol 3-galactoside and kaempferol 3-glucoside, which were not present in our samples. Catechin, epicatechin and procyanidins were present only in the leaves but not in the petals, as in the samples of *R. gallica* that we analyzed. In *R. subcanina*, we could not determine catechin and procyanidin dimer 3 and procyanidin trimer 1. Cunja et al. [20] found that quercetins accounted for 63.7% of flavonols. In our samples, these values were a little higher. In *R. gallica*, they accounted for 82% and in *R. subcanina,* 73% of the total flavonols. Schieber et al. [49] described that quercetin-3-rhamnoside was present in trace amounts in the petals of *R. damascena*. In our samples, it was the most abundant of the flavonols. Its contents were 0.36 g/kg FW in *R. gallica* and 0.25 g/kg FW in *R. subcanina*. Wan et al. [50] showed that kaempferol-3-O-rhamnoside was the major flavonol in the petals of *R. damascena* cultivars. Comparing the results of our two genotypes, it can be seen that we also identified the mentioned compound in our samples, but its content was not dominant. The content of the anthocyanin cyanidin-3-glucoside in our two genotypes was 0.18 g/kg FW (*R. gallica*) and 0.04 g/kg FW (*R. subcanina*). Considering the flower color, the low anthocyanin content in the petals of *R. subcanina* was expected. Kunc et al. [19] reported slightly higher values, 0.19 and 0.24 g/kg FW. Considering the anthocyanin content of other plants for comparison, Zannou et al. [51], for example, studied orange blossom (*Echium amoenum*), an annual herb native to the Mediterranean region, an excellent source of anthocyanins and used in various forms because of its biological activity. The dominant anthocyanin was cyanidin-3-glucoside, ranging from 0.03 to 1.13 g/kg. Li et al. [52] investigated the content of anthocyanins in 51 edible wildflowers. *Lilium brownii*, with a cyanidin-3-glucoside content of 0.106 mg/100 g, and *Ipomoea cairica* 0.109 g/kg, were highlighted as those with lower contents. The highest contents were recorded for *Jatropha integerrima* 6.41 g/kg and *Pelargonium hortorum* 4.97 g/kg.

## 4. Materials and Methods

### 4.1. Plant Material

Samples of rosehips and petals of *R. gallica* and *R. subcanina* (Figure 5) were collected in 2021 in the Podgorje area in the southwestern part of Slovenia. According to its morphological characteristics, *R. subcanina* lies between the related species *R. dumalis* Bechst. And *R. canina*. *R. gallica* thrives in all Slovenian regions, while *R. subcanina* occurs only in the northern and southeastern parts. *R. gallica* grows between 0.3 m and 1 m tall, *R. subcanina is* between 1.5 m and 2 m tall. The former thrives in sparse forests, barren meadows, pastures, and roadsides. It blooms from June to July and ripens in September. *R. subcanina* thrives on bushy slopes, forest edges and clearings. It flowers before *R. gallica*, from May to June, and ripens from September to October [2].

All rosehips were harvested at full maturity at stage BBCH 88, according to Meier et al. [53], and rose petals at stage BBCH 65, according to Meier et al. [54]. All plants from which we collected petals and hips grew under the same climatic conditions. The exact agrometeorological and pedological conditions have been previously described by Kunc et al. [1]. The collected material was placed on ice and taken to the laboratory, where ascorbic acid analyses were performed immediately. The material for other analyses was stored at −20 °C.

### 4.2. Extraction and Determination of Sugars and Organic Acids

The extraction of organic acids and sugars was performed and analyzed according to the method described by Mikulic-Petkovsek et al. [55]. Frozen rosehips (0.5 g), from which the seeds had been previously removed, were finely crushed, and 1.5 mL of bidistilled water was added. The samples in plastic centrifuge tubes were placed on a shaker (Unimax 1010, Heidolph Instruments, Hamburg, Germany) for half an hour. The extracts were centrifuged at 10,000× *g* for 7 min at 4 °C (Eppendorf Centrifuge 5810 R, Hamburg, Germany). The supernatant was filtered into a labeled vial through a syringe filter (Chromafil Xtra MV-20/25, Macherey Nagel, Dueren, Germany). Until further analysis, the samples were stored in a freezer at −20 °C. The extraction was performed in triplicate. We used a UV detector set to 210 nm for separating organic acids, a Rezex ROA column (Phenomenex, Torrance, CA, USA) heated to 65 °C, and a mobile phase containing 4 mM sulfuric acid at a flow rate of 0.6 mL/min. Sugars were also separated at 65 °C on a Rezex (Phenomenex, Torrance, CA, USA) RCM monosaccharide Ca + (2%) column (300 mm × 7.8 mm). The flow rate was 0.6 mL/min continuously, 30 min total run time, and bidistilled water as the mobile phase. Carbohydrates were detected using a refractive index (RI) detector.

### 4.3. Extraction and Determination of Ascorbic Acid

Ascorbic acid extraction was performed on rosehips, pulp with skin (without seeds), as described by Kunc et al. [29]. Fifteen mL of 3% meta-phosphoric acid was added to 0.5 g of material. Samples were shaken for 30 min at room temperature on a shaker platform (Unimax 1010, Heidolph Instruments, Schwabach, Germany) and then centrifuged at 10,000× *g* for 5 min at 4 °C (Eppendorf 5810 R Centrifuge, Hamburg, Germany). Samples were then filtered into vials through a Cromafil A-20/25 cellulose mixed ester filter (Macherey-Nagel, Dueren, Germany). The vials containing the extracts were stored at −20 °C until further analysis. Samples were analyzed using an HPLC system (Vanquish UHPLC, Thermo Fisher Scientific) and a UV detector set to 245 nm. The chromatographic conditions for determining ascorbic acid were previously described by Mikulic-Petkovsek et al. [56]. A Rezex ROA column (Phenomenex, Torrance, CA, USA) was used to separate ascorbic acid. It was operated at 20 °C with 4 mM sulfuric acid as the mobile phase.

### 4.4. Extraction and Determination of Carotenoids

Carotenoids were extracted according to the method described by Mikulic-Petkovsek et al. [56]. Briefly, 0.2 g of the frozen material (pulp with skin, without seeds) was extracted in glass centrifuges with 2 mL of acetone at a temperature of 4 °C using an Ultra-Turrax (IKA-Werke GmbH & Co. KG, Staufen, Germany) homogenizer for 30 s and determined on the Accela HPLC system (Thermo Scientific, San Jose, CA, USA) using the gradient method. Samples were then filtered into labelled vials through a Cromafil A-20/25 polyamide/nylon filter (Macherey-Nagel, Dueren, Germany). The vials containing the extracts were stored at −20 °C until further analysis. The extracts were then analyzed using HPLC-DAD (Thermo Finnigan, San Jose, CA, USA) at 450 nm with a Gemini C18 column (150 × 4.6 mm 3 μm; Phenomenex, Torrance, CA, USA). The first mobile phase was solvent A: acetonitrile/methanol/water (100/10/5, *v*/*v*/*v*) and the second solvent B: acetone/ethyl acetate (2/1, *v*/*v*). The flow rate was 1 mL/min with the following gradient: from 10 to 70% B in the first 18 min, then linearly at 70% B up to 22 min and back to the initial conditions until the end of the run.

### 4.5. Extraction and Determination of Phenolic Compounds

Phenolic compounds were studied from the petals of the investigated genotypes. Extractions were performed according to the extraction method previously described by Kunc et al. [19]. All analyses were performed in triplicate. Each sample was individually crushed in a mortar with liquid nitrogen, and an accurately measured mass of the sample was placed in a centrifuge tube, to which the extraction solution, 3% formic acid in methanol with bidistilled water, was added. The ratio of weighed sample to the extraction solution was always 1:5. The mass of all flower samples was 0.02 g, and the volume of the extraction solution was 1 mL. Extraction took place for 1 h in a cooled ultrasonic bath (Iskra PIO, SONIS 4 GT, (Iskra PIO d.o.o., Šentjernej, Slovenia)), and the extract was then centrifuged at 10,000× *g* for 7 min at 4 °C with an Eppen-Dorf 5810 R centrifuge. The supernatant was filtered through a 0.20 mm polyamide/nylon filter (Macherey-Nagel, Dueren, Germany). Vials with extracts were stored at −20 °C until further analysis of phenolic compounds.

Analysis of phenolic constituents was performed using the Thermo Scientific Dionex HPLC system with a diode detector (Thermo Scientific, San Jose, CA, USA) connected to Chromeleon workstation (software Launch Chromeleon 7). The chromatographic method for phenol analysis was previously described by Mikulic-Petkovsek et al. [56]. The detector was set to three wavelengths: 280 nm, 350 nm, and 530 nm. The mobile phases used were A: 3% acetonitrile/0.1% formic acid/96.9% bidistilled water; phase B: 3% water/0.1% formic acid/96.9% acetonitrile. Gradient elution of both mobile phases is described in Mikulic-Petkovsek et al. [57], and the flow rate was 0.6 mL/min. The column was a Gemini C18 (150 × 4.6 mm 3 μm; Phenomenex, Torrance, USA) heated to 25 °C.

Phenolic compounds were identified using electrospray ionization (ESI) mass spectrometer (LTQ XL Linear Ion Trap Mass Spectrometer, Thermo Fisher Scientific, Torrance, CA, USA). It works in positive (anthocyanins) or negative (all other phenolis) ionization mode. All mass spectrometer conditions were identical to those described by Mikulic-Petkovsek et al. [57]. Spectral data were generated using Excalibur (Thermo Scientific, Torrance, CA, USA). Identification of the compounds was confirmed by comparison of retention times and their spectra, the addition of standard solution to the sample, and fragmentation and comparison of the obtained results with literature data.

### 4.6. Statistical Analysis

For statistical data processing, data were collected using Microsoft Excel 2016 and R Commander (R i386 4.1.2) using one-way analysis of variance (ANOVA) with species as a factor. Duncan’s test was used to compare treatments when ANOVA showed significant differences between values. Results were expressed as mean ± standard error (SE). When p-values were less than 0.05, differences between genotypes were statistically significant. Principal component analysis (PCA) was performed for all rosehip data (pulp with skin). We also checked the correlation of each metabolite with the Pearson correlation with the corresponding matrix.

## 5. Conclusions

We found a strong positive correlation between the total content of phenolic compounds and ascorbic acid, the total content of organic acids and the total content of carotenoids, and the total content of sugars and organic acids. The studied rosehips had an extremely low sugar content and, consequently, an extremely high organic acid content. The carotenoid content of the rosehips was in the lower range of the average content previously reported in the literature. However, since the carotenoid content depends, among other things, on the harvesting time, it would be necessary to study different harvesting dates in more detail to check whether the carotenoid content is higher on a specific date. Rose hips with higher carotenoid content, by optimizing the harvesting time, could potentially be used for industrial purposes. In addition, how the content of ẞ-carotene in rosehips changes, which is necessary for the formation of vitamin A, would also be interesting to analyze in the future. However, we found that the analyzed flowers are a rich source of phenolic compounds that benefit humans and could potentially be used in the food and cosmetic industries.

## Figures and Tables

**Figure 1 plants-12-02979-f001:**
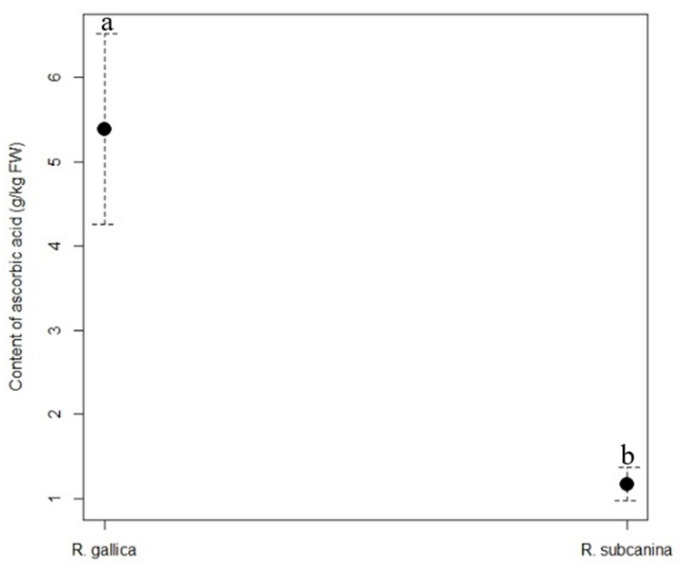
Content (g/kg FW, mean ± SE) of ascorbic acid in hips of *R. gallica* and *R. subcanina*, collected in Podgorje. Different letters indicate statistical differences between genotypes.

**Figure 2 plants-12-02979-f002:**
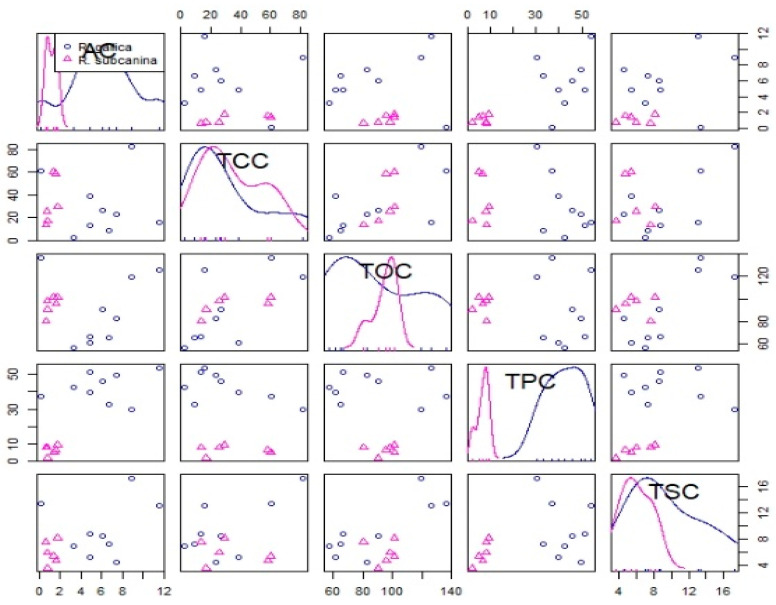
Correlation test of all analyzed metabolites of *R. gallica* (circles) and *R. subcanina* (triangles) samples.

**Figure 3 plants-12-02979-f003:**
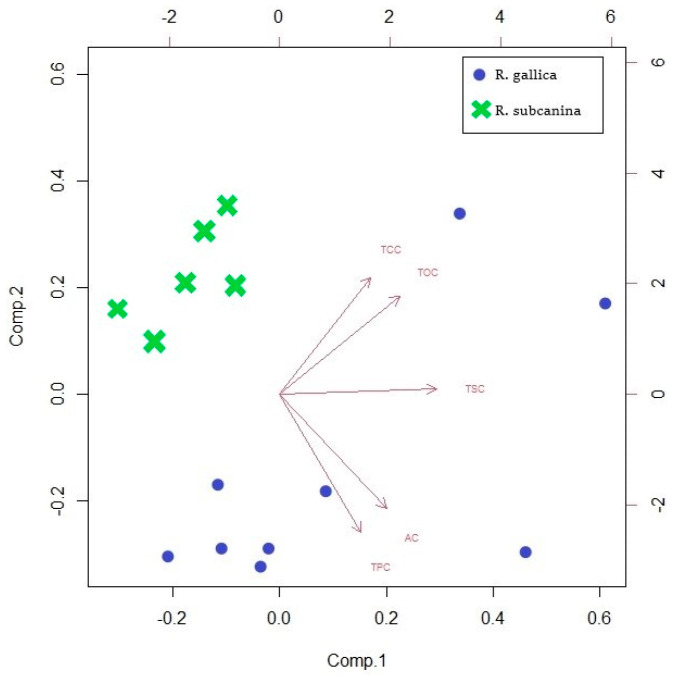
Biplot corresponding to PCA for all analyzed samples.

**Figure 4 plants-12-02979-f004:**
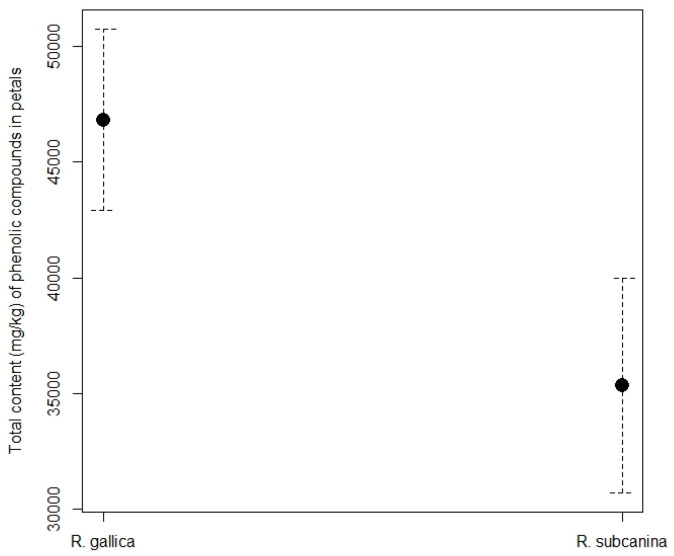
Total content (mg/kg FW, mean ± SE) of phenolic compounds in petals of *R. gallica* and *R. subcanina*, collected in Podgorje.

**Figure 5 plants-12-02979-f005:**
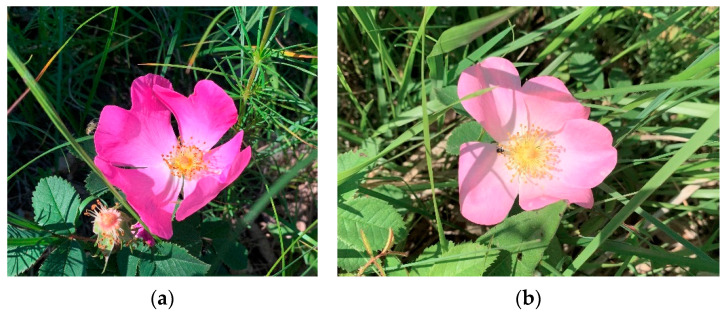
(**a**) Petals of *R. gallica* and (**b**) *R. subcanina.*

**Table 1 plants-12-02979-t001:** Content (g/kg FW, mean ± SE) of sugars in hips of *R. gallica* and *R. subcanina*, collected in Podgorje. Different letters indicate statistical differences between genotypes.

Sugar	*R. gallica*	*R. subcanina*
Sucrose	2.98 ± 0.6 a	1.42 ± 0.3 a
Glucose	2.83 ± 0.4 a	2.19 ± 0.2 a
Fructose	3.36 ± 0.6 a	2.34 ± 0.5 a
Mannitol	0.29 ± 0.05 a	0.17 ± 0.03 a
Sorbitol	0.03 ± 0.01	-
TOTAL	9.49 ± 1.66 a	6.12 ± 1.03 a

(-): Compound was not detected.

**Table 2 plants-12-02979-t002:** Content (g/kg FW, mean ± SE) of organic acids in hips of *R. gallica* and *R. subcanina*, collected in Podgorje. Different letters indicate statistical differences between genotypes.

Organic Acid	*R. gallica*	*R. subcanina*
citric acid	18.29 ± 1.67 b	39.34 ± 8.34 a
malic acid	7.89 ± 1.61 b	18.86 ± 1.06 a
quinic acid	44.07 ± 3.88 a	55.11 ± 3.61 a
shikimic acid	0.14 ± 0.02 a	0.19 ± 0.04 a
fumaric acid	0.04 ± 0.003 b	0.048 ± 0.01 a
TOTAL	70.39 ± 5.44 b	113.57 ± 6.65 a

**Table 3 plants-12-02979-t003:** Content (mg/100 g FW, mean ± SE) of carotenoids in hips of *R. gallica* and *R. subcanina*, collected in Podgorje. Different lowercase letters indicate statistically significant differences between genotypes.

	Lutein	Zeaxanthin	Lycopene	α-Carotene	ẞ-Carotene	TOTAL
*R. gallica*	0.74 ± 0.17 a	0.29 ± 0.09 a	2.19 ± 0.39 a	1.59 ± 0.76 a	27.56 ± 6.43 a	32.39 mg/100 g
*R. subcanina*	2.09 ± 1.29 a	0.33 ± 0.06 a	2.21 ± 0.12 a	0.27 ± 0.12 a	23.37 ± 6.85 a	28.27 mg/100 g

**Table 4 plants-12-02979-t004:** Content (mg/kg FW, mean ± SE) of the main phenolic groups in pulp with skin and seeds in hips of *R. gallica* and *R. subcanina*, collected in Podgorje. Different letters indicate statistically significant differences between genotypes (separated pulp with skin and seeds).

Phenolic Group	*R. subcanina* Pulp with Skin	Seeds	*R. gallica* Pulp with Skin	Seeds
Hydroxybenzoic acid derivatives (HBA)	12.80 ± 6.28 b	72.68 ± 28.06 a	142.27 ± 40.10 a	3.76 ± 1.65 b
Hydroxycinnamic acid derivatives (HCA)	202.30 ± 124.65 a	328.69 ± 102.55 a	147.22 ± 36.17 b	1.58 ± 0.44 b
Gallotannins	912.96 ± 456.91 a	282.94 ± 188.93 a	761.48 ± 262.74 b	221.99 ± 95.90 a
Ellagitannins	503.40 ± 223.61 a	72.98 ± 26.91 a	343.60 ± 115.04 b	76.10 ± 49.97 a
Flavanols	3608.40 ± 1061.89 b	483.07 ± 294.17 a	12,293.14 ± 3132.78 a	295.23 ± 122.32 b
Flavonols	55.69 ± 18.05 b	12.63 ± 6.09 a	157.94 ± 35.58 a	14.49 ± 6.49 a
Flavones	8.77 ± 0.82 a	-	0.12 ± 0.02 b	-
Dihydrochalcone	-	10.09 ± 4.38 b	-	1098.45 ± 544.72 a
Cyanidin-3-glucoside	1.13 ± 0.45 b	-	-	28.78 ± 5.91 a
TOTAL	5305.45 ± 1892.66 b	1263.08 ± 651.09 b	15,767.21 ± 3628.34 a	1711.60 ± 821.49 a

(-): Compound was not detected. Consideration of the content higher than 0.001 mg/kg.

**Table 5 plants-12-02979-t005:** Pearson correlations for analyzed metabolites in pulp with skin: ascorbic acid (AC), total carotenoids content (TCC), total organic acids content (TOC), total phenolics content (TPC) and total sugars content (TSC).

	AC	TCC	TOC	TPC	TSC
**AC**	1.00	−0.07 NS	0.05 NS	0.72 *	0.46 NS
**TCC**	−0.07	1.00	0.60 *	−0.24 NS	0.41 NS
**TOC**	0.05	0.60	1.00	−0.13 NS	0.61 NS
**TPC**	0.72	−0.24	−0.13	1.00	0.38 NS
**TSC**	0.46	0.41	0.61	0.38	1.00

Significance codes: ‘*’ = 0.05; NS ‘no statistical difference’.

**Table 6 plants-12-02979-t006:** Content ± standard error (mg/kg FW) of phenolic compounds (HCA, gallotannins and ellagitannins) in *R. gallica* and *R. subcanina petals*. Different letters indicate significant differences between genotypes.

Phenolic Group	Compound	*R. gallica*	*R. subcanina*
Hydroxycinnamic acid derivatives (HCA)	*p*-coumaric acid hexoside 1	4.85 ± 1.02 a	0.62 ± 0.19 b
	5-*p*-coumaroylquinic acid 1	38.41 ± 3.02 a	18.75 ± 1.82 b
	5-*p*-coumaroylquinic acid 2	5.75 ± 0.81 a	0.19 ± 0.02 b
	3-*p*-coumaroylquinic acid	5.52 ± 1.48 a	-
	3-caffeoylquinic acid	3.79 ± 0.98	-
	4-caffeoylquinic acid	3.91 ± 0.83	-
	5-caffeoylquinic acid 1	4.97 ± 0.92	-
	5-caffeoylquinic acid 2	6.67 ± 0.71	-
	TOTAL	73.87 ± 9.77 a	19.56 ± 2.03 b
Gallotannins	Trigalloyl hexoside 1	18,831.47 ± 3032.47 a	28,515.69 ± 4702.61a
	Trigalloylhexoside 2	6.90 ± 1.72 a	2.04 ± 0.72 a
	Trigalloylhexoside 3	-	1.61 ± 0.11
	TOTAL	18,838.37 ± 3034.19 b	28,519.34 ± 4703.44 a
Ellagitannins	HHDP digalloylhexoside isomer 1	0.01 ± 0.0 a	0.02 ± 0.0 a
	HHDP digalloylhexoside isomer 2	369.28 ± 62.33 a	165.21 ± 55.52 b
	HHDP digalloylhexoside isomer 3	508.83 ± 112.20	-
	Galloyl bis HHDP hexoside 1	6.17 ± 0.88	-
	TOTAL	884.29 ± 175.41 a	165.23 ± 55.52 b

(-): Compound was not detected. Consideration of the content higher than 0.001 mg/kg.

**Table 7 plants-12-02979-t007:** Contents ± standard error (mg/kg FW) of flavanols, flavonols and dihydrochalcone in petals of *R. gallica* and *R. subcanina*. Different letters indicate significant differences between genotypes.

Phenolic Group	Compound	*R. gallica*	*R. subcanina*
Flavanols	Dimer PA monogallate 1	900.27 ± 70.81 a	600.67 ± 86.34 b
	Dimer PA monogallate 2	787.47 ± 57.20	-
	Procyanidin dimer 1	242.13 ± 26.13 b	369.47 ± 72.28 a
	Procyanidin dimer 2	192.99 ± 52.05 a	115.84 ± 19.10 b
	Procyanidin dimer 3	897.13 ± 221.43	-
	Catechin	184.98 ± 47.38	-
	Epicatechin	87.01 ± 9.46 a	74.09 ± 22.51 b
	Procyanidin trimer 1	274.72 ± 39.21	-
	TOTAL	3566.61 ± 523.67 a	1160.07 ± 200.23 b
Flavonols	Quercetin dihexoside 1	4836.67 ± 966.60	-
	Quercetin dihexoside 2	6624.44 ± 873.07	-
	Galloyl hexoside	1191.72 ± 316.71	-
	Trigalloyl HHDP hexoside	1087.15 ± 324.07	-
	Kaempferol dihexoside 1	0.16 ± 0.12	-
	Kaempferol dihexoside 2	0.51 ± 0.08	-
	Quercetin-3-rutinoside	84.40 ± 16.79 a	39.77 ± 34.00 b
	Quercetin galloyl hexoside 1	633.62 ± 74.42 a	283.86 ± 38.79 b
	Quercetin galloyl hexoside 2	339.14 ± 64.54 a	233.79 ± 21.44 b
	Quercetin pentosyl hexoside	969.17 ± 249.47	-
	Quercetin-3-galactoside	898.47 ± 138.21 a	212.51 ± 177.60 b
	Quercetin-3-glucoside	220.00 ± 19.44 a	147.5 ± 22.26 b
	Quercetin-3-xyloside	20.22 ± 2.79 a	20.39 ± 3.74 a
	Kaempferol hexoside 1	52.67 ± 11.53 a	33.00 ± 5.32 b
	Kaempferol hexoside 2	852.11 ± 181.44 a	553.75 ± 77.64 b
	Quercetin-3-arabinopyranoside	983.33 ± 138.42 a	180.01 ± 20.53 b
	Quercetin-3-arabinofuranoside	1584.98 ± 326.04 a	2541.73± 149.79 b
	Quercetin acetylhexoside	1.58 ± 0.32 b	4.45 ± 1.17 a
	Quercetin-3-rhamnoside	358.24 ± 37.53 a	250.06 ± 55.68 b
	Kaempferol-3-glucuronide	163.21 ± 77.73 b	266.25 ± 44.06 a
	Quercetin galloylpentoside 1	29.78 ± 11.46	-
	Quercetin galloylpentoside 2	161.33 ± 57.62	-
	Kaempferol pentoside 1	67.82 ± 19.96 a	58.75 ± 13.65 a
	Kaempferol pentoside 2	589.83 ± 47.11 a	123.88 ± 29.56 b
	Kaempferol pentoside hexoside	168.33 ± 70.40	-
	Kaempferol rutinoside	36.63 ± 7.67 b	242.5 ± 60.32 a
	Kaempferol acetylhexoside	46.56 ± 3.57 a	40.75 ± 10.81 a
	Kaempferol rhamnoside	258.88 ± 48.43 b	376.25 ± 90.95 a
	TOTAL	22,260.95 ± 4085.54 a	5609.20 ± 857.31 b
Dihydrochalcone	phloridzin	1084.44 ± 142.84	-

(-): Compound was not detected. A content higher than 0.001 mg/kg was considered.

**Table 8 plants-12-02979-t008:** Average content (mg/kg FW, mean ± SE) of cyanidin-3-glucoside in petals of *R. gallica* and *R. subcanina*, collected in Podgorje. Different lowercase letters indicate statistically significant differences between genotypes.

Phenolic Group	Compound	*R. gallica*	*R. subcanina*
Anthocyanins	cyanidin-3-glucoside	182.89 ± 73.12 a	40.62 ± 5.21 b

## Data Availability

All data are presented within the article.

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
