# Peer review of "Detailed Metabolic Characterization of Flowers and Hips of Rosa gallica L. Grown in Open Nature"

_plants, 2023, doi:10.3390/plants12162979_

Round 1

Reviewer 1 Report

Line 12 – „In the study, the content and composition of various bioactive compounds (sugars …..)” – Sucrose, glucose and fructose determined in this study are not bioactive compounds.

Lines 117-118 – „2.1. Bioactive compounds in rose hips. The total sugar content (sucrose, glucose, fructose ….)” – These sugars are not bioactive compounds.

Lines 152-153 – „For all analyzed carotenoids (Table 3), there were no statistically significant differences between the genotypes studied” and lines 156-158 – „There was no statistically significant difference between the two samples in the content of analyzed carotenoids” – Information in these sentences is repeated.

Line 155 –  It is 23.27 in the text, and in Table 3 it is 23.37.

Line 174 – replace flavonols with flavanols

Table 4 – This table shows the results of cyanidin-3-glucoside content, and in the text of the manuscript (section 2.1) does not describe these results.

Line 236-237 – „There is a statistically significant difference between the total content of flavanols, flavonols, and dihydrochalcones in rose petals (Table 7)” – Dihydrochalcones were detected only in R. gallica.

Line 241 – In the text is 5609.27, and in Table 7 is 5609.20

Line 245 – delete leaves

Line 351 – delete [1]

Line 362 – The name gallatonins is incorrect, replace it with gallotannins.

Line 364 - The name gallatonins is incorrect, replace it with gallotannins.

Line 381 – Instead of flavanols should be flavonols.

Line 385 – Instead of flavanol it should be flavonol.

Line 430 – Insert city, country after Eppendorf Centrifuge 5810 R.

Lines 458-459 – insert manufacturer, city and country after Ultra-Turrax homogenizer

Lines 479 – Insert city and country after Iskra PIO, SONIS 4 GT.

Line 579 – No volume number, no page range given.

Lines 589, 608, 626-627, 633, 639, 649, 655, 667-668 – According to Instructions for Authors, abbreviations of journal names should be given.

Line 599 – Rosa rugosa - Latin name should be written in italics.

Line 600 – Give the name of the journal, the year, the volume number according to the Instructions for Authors (J. Food Qual. – italic, year – bold, volume number – italic)

Line 652 – Article title not given.

Line 661 – Year 2013 is incorrect, page numbers are missing.

Reviewer 2 Report

I think that the data are important for overall knowledge about the different Rosa species chemical composition and their potential use as food or medicines.

I think the authors does not have a special aim of this work, or I can not find in abstract and generally in paper. And this have negative impact on overall presentation.

In introduction there are a lot of data presented generally, like some kind of substances were already studied in these species. I think it must be more specific, being a literature data.

Some results are expressed in mg/kg, that is an unusual UM. In all paper are results presented in different UM and can not be easily compared. I suggest to unify the UM through all paper.

At figure 4 can not be made easily difference between the two species. I suggest one of species to receive a different mark in place of bullet.

At all bioactive compounds class the authors expressed the total content, but this is not very correct, because it is only the total of individual compounds determined from that class. For this reason some comparison with literature data are not clear, because by the used methods can not be with certitude determine all bioactive compounds from a plant material.

At discussions a lot of results are repeated. At sugars discussion the comparisons are wrong, the conclusion are not sustained by the given data. E.g. the authors determined 9.49 g/kg sugars and only fructose content in literature data is 13,58 mg/g meaning 13,58 g/kg. The conclusion of authors is that their samples have higher sugar content that the samples with that they compare them.

At polyphenols the total content expressed in mg GAE/100 g can not be compared with the total content determined by authors. The reason is that the authors determined a number of polyphenols, but not all with certitude and the total content given by them is the sum of identified compound.

I can not find a conclusion of comparisons from discussions, into the discussions. For me it seems to be a list of data, without a really discussion.

At extraction for sugars and organic acids why is made a centrifugation before the extraction? There is not explained without doubt which are the samples solutions that after were analysed.

At LC/MS method spectral data which kind of data were taken from literature?

I think for all these observations the paper needs a major revision.

The language is poor to medium, sometimes is difficult to understood. I suggest to be checked by a people who is well-speaking English.

Reviewer 3 Report

This paper provides a detailed report on the content of several compounds in two rose species naturally found in Podgorje. The experiments were soundly performed and the results can be of interest for further use of these plants.

The data is not always well presented, and some parts of the manuscript are not clear enough. In addition, improvements should be made on the English syntax. Specific comments are given in the attached file, but the whole paper should be edited again. 

In general, the reading would be more fluent and powerful if some of the numbers are removed from the discussion part, since they appear in detail in the Results section.

Round 2

Reviewer 3 Report

accepted